# Lignin as a Bioactive Additive in Chlorzoxazone-Loaded Pharmaceutical Tablets

**DOI:** 10.3390/molecules30071426

**Published:** 2025-03-23

**Authors:** Andreea Creteanu, Gabriela Lisa, Cornelia Vasile, Maria-Cristina Popescu, Daniela Pamfil, Alina-Diana Panainte, Gladiola Tantaru, Madalina-Alexandra Vlad, Claudiu N. Lungu

**Affiliations:** 1Department of Pharmaceutical Technology, Faculty of Pharmacy, “Grigore T. Popa” University of Medicine and Pharmacy, 16 University Street, 700115 Iași, Romania; acreteanu@gmail.co; 2Department of Chemical Engineering, Faculty of Chemical Engineering and Environmental Protection, “Gheorghe Asachi” Technical University, 73 Prof. Dimitrie Mangeron, Street, 700050 Iași, Romania; gapreot@ch.tuiasi.ro; 3Physical Chemistry of Polymers Department, Petru Poni Institute of Macromolecular Chemistry, 41A Gr. Ghica Voda Alley, 700487 Iași, Romania; cvasile@icmpp.ro (C.V.); cpopescu@icmpp.ro (M.-C.P.); pamfil.daniela@icmpp.ro (D.P.); 4Department of Analytical Chemistry, Faculty of Pharmacy, “Grigore T. Popa” University of Medicine and Pharmacy, 16 University Street, 700115 Iași, Romania; gtantaru2@yahoo.com; 5Department of Microbiology, Faculty of Pharmacy, “Grigore T. Popa” University of Medicine and Pharmacy, 16 Universitatii Street, 700115 Iași, Romania; madalina.vlad@umfiasi.ro; 6Department of Functional and Morphological Science, Faculty of Medicine and Pharmacy, Dunarea de Jos University, 800008 Galati, Romania

**Keywords:** chlorzoxazone, lignin, matrix tablets, antioxidant, antimicrobial properties

## Abstract

In the present work, the application of lignin (LIG) as a bioactive additive for the preparation of drug-loaded tablets by direct compression has been studied, and its influence on the release of chlorzoxazone (CLZ) from the hydrophilic matrices has been followed. In hydrophilic matrices, the excipients Kollidon^®^ SR (KOL) and chitosan (CHT) have been used in various amounts and tested in the preparation of 500 mg tablets. They were used as matrix-forming agents, and their influence on the flow and the compressibility properties as well as their effect on the pharmaco-chemical characteristics of the matrix tablets have been studied. Based on the initial evaluation of the pharmaco-technical analysis, pharmaco-chemical characteristics, and in vitro release profile, three matrix tablet formulations (FLa, FLb, and FLc) were selected and further tested. They were evaluated through Fourier-transform infrared spectrometry (FTIR), X-ray diffraction (XRD), thermogravimetry (TG), differential scanning calorimetry (DSC), and in vitro dissolution tests. The three formulations were comparatively studied regarding the release kinetics of active substances using in vitro release testing. The in vitro kinetic study reveals a complex release mechanism occurring in two steps of drug release. The first one is a burst effect that occurs within the first 0–2 h, involving a rapid release of the majority of the drug in a short time, followed by the second step as a prolonged release of the drug, which is relatively constant with a fixed rate over the next 2–36 h. Two factors have been calculated to assess the release profile of chlorzoxazone: f1—the similarity factor and f2—the difference factor together with the correlation coefficient R^2^. Comparing their values, the three optimal formulations have been selected, containing 55 mg LIG (FLa), 60 mg LIG (FLb), or 65 mg LIG (FLc), confirming that LIG next to KOL and CHT influenced the release characteristics of the matrix tablets. Due to the presence of lignin in the matrix of the three formulations, FLa, FLb, and FLc tablets with CLZ, the antioxidant activity has improved. The antioxidant activity of FLc was found to be 21.36% ± 1.06 greater than that of FLa and FLb. The tablets FLa, FLb, and FLc also presented higher antimicrobial activity against *Staphylococcus aureus*, *Escherichia coli*, *Candida albicans*, and colistin-resistant *Klebsiella* spp. The higher the concentration of LIG in the matrix (FLc), the higher the antimicrobial activity. By using LIG, the drug dose could be decreased. It can be concluded that lignin can be used as a multifunctional pharmaceutical bioactive additive/excipient for tablets. Its interesting properties have been proven, and its use as a pharmaceutical active additive should be exploited for different applications.

## 1. Introduction

Tablets are the most commonly used pharmaceutical oral dosage form [1]. They are relatively simple to manufacture, show good physical stability, and are extensively accepted by patients [1,2]. They are typically composed of several small amounts of active pharmaceutical ingredients and, to the greatest extent, of excipients that contribute to the preservation of the tablet’s structure and stability, as well as, ideally, provide drug delivery and/or protection function after ingestion. The clinical and formulation performances of pharmaceutical forms (tablets, capsules, suspensions, etc.) depend on the physicochemical properties of the excipients [3]. Biopolymers are being used for this purpose since they not only meet the need of the industry to develop efficient oral dosage forms with good solubility, release, and biocompatibility but also have low toxicity, biodegradability, and low price [4].

Biomolecules like inulin, chitin, starch chitosan, pectin, cellulose, etc. are being used as excipients for the delivery and controlled delivery of active ingredients, although they are typically high in price [5]. Drug loading into a matrix system is the most popular way to prevent its release [6]. Different pharmaceutical excipients can be used for direct compression, including a wide range of polymers [7]. These polymers include synthetic macromolecules, such as poly (vinyl pyrrolidone) or poly (acrylic acid), and natural polymers, such as cellulose [8].

Chlorzoxazone (5-chloro-2,3-dihydro-1,3-benzoxazol-2-one) (CLZ) is considered a Class II drug (according to the Biopharmaceutical Classification System (BCS)) because it has low solubility and high membrane permeability. CLZ has been licensed for treating musculoskeletal disorders [9] and is used to relieve muscle spasms and subsequent pain and discomfort [10]. The recommended starting oral dosage is 500 mg three or four times per day; however, this can frequently be lowered to 250 mg three or four times per day in the future. CLZ is typically administered alongside analgesics. The three most typical side effects of CLZ are headache, lightheadedness, and drowsiness [11]. Rarely, there have been reports of severe—even fatal—hepatocellular toxicity with CLZ treatment. Early hepatotoxicity signs and/or symptoms, such as fever, rash, anorexia, nausea, vomiting, lethargy, right upper quadrant pain, dark urine, or jaundice, should be reported by patients [12]. Chlorzoxazone is rapidly absorbed from the gastrointestinal tract. Therapeutically active plasma concentrations are maintained for 3–4 h.

Water-insoluble medicines have limited therapeutic effects due to their low solubility and dissolving rates [13]. The gastrointestinal absorption of any active substance, such as amiodarone, is significantly reduced due to its low water solubility.

In earlier research, we examined various methods to improve the active ingredient’s solubility (amiodarone) and the impact of formulation parameters on CLZ’s stability [14,15,16].

The study is based on novel oral matrix tablets designed to maximize the low oral bioavailability using lignin, chitosan, and Kollidon^®^ SR.

Lignin is one of the most abundant polymers on Earth, second after cellulose [17]. Lignin is a complex organic polymer found in the cell walls of plants, particularly in wood and bark. The potential applicability of lignin is aimed at biomedical applications, including tissue engineering, wound dressings, and pharmaceutical uses, i.e., excipients and drug delivery systems [18].

However, the use of LIG as an excipient for pharmaceutical formulations is scarce, and only a few studies describe its use [18]. This biopolymer has been used in a wide variety of applications, such as antimicrobial agents, antioxidant additives, and UV protective agents. Lignin has significant antioxidant properties due to its phenolic structure, which allows it to scavenge free radicals. This activity is beneficial in reducing oxidative stress in biological systems and can be harnessed in the development of health supplements and pharmaceuticals [19].

Some studies suggest that lignin and its derivatives can reduce inflammation by modulating the activity of inflammatory mediators. Certain lignin derivatives have been found to exhibit cytotoxic effects on cancer cells, inhibiting their growth and proliferation. This anticancer potential is an ongoing area of research, with the goal of developing lignin-based therapeutic agents [20]. Due to its biocompatibility and biodegradability, lignin is being explored as a carrier material for drug delivery systems [21].

Lignin nanoparticles can encapsulate drugs, enhancing their stability and controlled release. Lignin can act as a prebiotic, promoting the growth of beneficial gut bacteria. This property is essential for maintaining a healthy digestive system and could be utilized in the development of functional foods and dietary supplements. Lignin can inhibit key enzymes involved in microbial metabolism.

The phenolic compounds in lignin can bind to the active sites of these enzymes, preventing them from catalyzing essential biochemical reactions [22]. Lignin and its derivatives can interact with microbial DNA, causing structural changes or damage. This interaction can inhibit DNA replication and transcription, thereby preventing microbial proliferation. Lignin can chelate essential metal ions such as iron and magnesium, which are necessary for microbial growth and enzyme function. By sequestering these ions, lignin deprives microorganisms of critical nutrients, inhibiting their growth [23]. Lignin can act synergistically with other antimicrobial agents, enhancing their efficacy. This synergy can occur through various mechanisms, such as disrupting microbial defenses or facilitating the entry of other antimicrobials [24].

Kollidon^®^ SR (Crospovidone) (KOL) is a physical mixture of polymers. KOL is one of the utilized hydrophilic excipients in the formulation and production of modified-release matrix tablets [25]. Kollidon^®^ SR forms chemical reversible complexes or associates with a large number of drugs. The formation of the complexes with a drug depends very much on its chemical structure. The ability to form complexes has many uses in pharmaceuticals, including the following:Improving the dissolution and bioavailability of drugs;Adsorbing and removing polyphenols and tannins from tinctures and herbal extracts and improving the taste of azithromycin, paracetamol, and vitamins [26].

Chitosan (CHT) is a heteropolysaccharide formed by deacetylation of chitin, an abundant biopolymer extracted from insects/crustacean shells/mushroom cell walls. It possesses bioadhesive, wound-healing, and film-forming characteristics [27].

CHT is a biodegradable and biocompatible polymer that acts as an absorption promoter for hydrophobic active substances with high molecular weight in the gastrointestinal tract [28].

Other materials used in the present study were microcrystalline cellulose as Avicel^®^ PH (AV) and magnesium stearate (ST). All used compounds accomplish the quality requirements according to the laws of force. These excipients facilitate the application of the direct compression method to obtain optimal dispersibility of the hydrophobic substances (CLZ).

In the present work, we proposed the use of LIG as an excipient for direct compression in the preparation of drug-containing tablets and selected Chlorzoxazone. Additionally, LIG was combined with KOL, CHT, AV, and ST to prepare different types of tablets. In previous studies, the optimal amount of KOL was 30% (*w*/*w* %) and CHT was 5% (*w*/*w* %) for the hydrophilic matrix of tablets with CLZ [29]. In the new tablet formulations containing CLZ, the hydrophilic matrix maintained identical amounts of KOL and CHT excipients, while the quantity of LIG was varied and pressure was applied to obtain the tablets. In the studies that were conducted, the influence of LIG on the release of CLZ from the hydrophilic matrices and the antioxidant and antimicrobial actions of the tablets due to LIG presence and quantity were monitored. The new tablets were characterized by evaluating their pharmaco-technical parameter of the matrix, drug-excipients compatibility, in vitro dissolution tests, and drug release kinetics, as well as their antioxidant and antimicrobial properties.

## 2. Results

### 2.1. Pharmaco-Technical Parameter Values of Matrix Tablet Formulations

The results obtained for the pharmaco-technical parameters of matrix tablet formulations are shown in Table 1.

Formulations Fla, FLb, and FLc showed variations in tablet mass within the limits set by European Pharmacopoeia, 11^th^ ed. [30].

The analysis of the obtained values proved that the FLc formulations containing the highest percentage of LIG exercise the slightest variation in mass uniformity, below the 5% limit set by European Pharmacopoeia, 11^th^ ed. [30].

Tablets prepared using higher amounts of LIG showed lower thickness and friability. However, the differences in the thickness of tablets made with less than 13 (*w*/*w* %) of LIG end 90 kN compression force (Table 1).

Mechanical strength varied between 85.73 N and 89.62 N, and the values decreased directly proportionally to the increase in the concentration of LIG (Table 1).

When LIG/KOL and CHT blends were used as excipients, they showed higher crushing strength (resistance to crushing) values (Table 1).

The mass uniformity showed a deviation of ± 5% compared to the mass of standard tablets of ≥250 mg (500 ± 25 mg).

The dose uniformity showed a deviation of ± 5% compared to the mass of standard tablets of ≥ 250 mg (500 ± 25 mg). The dose uniformity values corresponded to each unit’s content of active substance, which should be 95% to 105% as the average content.

The data in Table 1 show that the average drug content of three tablets from each formula was 248 ± 0.912 mg for Fla, 249 ± 0.963 mg for FLb, and 250 ± 0.028 mg for FLc. The standard deviation (SD) values indicated that the sample processing performed did not imply significant modifications.

CLZ was quantitatively determined using an HPLC method. The chromatograms of CLZ pure substance (25 *w*/*w* %) and CLZ from the formulations (Fla, FLb, FLc 50 *w*/*w* %) are given in Figure 1. The detection of CLZ was performed at its characteristic wavelength of 280 nm (Figure 1).

From Figure 1, it was observed that the shape and intensity of the peak corresponding to CLZ in the formulations did not change because the drug quantity was constantly maintained throughout all formulations. LIG from the matrix was also identified using an HPLC method with UV detection at 216 nm (Figure 2). The corresponding LIG chromatograms from the three formulations (Figure 2) highlighted intensity variations that were proportional to LIG content (Table 1).

Friability, a pharmaco-technical parameter directly correlated with the mechanical strength of the tablets, showed increasing values directly proportional to the increased concentration of LIG in the formulations (Table 1).

Figure 3 illustrates the hydration characteristics of the hydrophilic matrix tablets in relation to CLZ concentration. The matrix-forming polymers directly influenced the hydration parameters of the hydrophilic matrix tablets with modified release.

Thus, Fla, FLb, and FLc formulations with 30 *w*/*w* % KOL, 5 *w*/*w* % CHT, and 11–13 *w*/*w* % LIG concentration exhibited absorbent properties in the first six hours of the test; after that, in the following 2–3 h, there was a slight decrease in mass. After 10 h, the matrix tablets were almost unchanged in size and mass (Figure 3). It is worth mentioning that formulation FLc with 30 *w*/*w* % KOL, 5 *w*/*w* % CHT, and 13 *w*/*w* % LIG showed superior hydration properties to other combinations. Formulations Fla and FLb, with 30 *w*/*w* % KOL, 5 *w*/*w* % CHT, and 11–12 *w*/*w* % LIG, were characterized by inferior adsorbent properties.

Therefore, LIG controlled the hydration degree of the tablets, and it also presented antioxidant and antimicrobial activities that would add value to the final product [31] (see below).

### 2.2. Drug-Excipients Compatibility Study

#### 2.2.1. FTIR Spectroscopy

Figure 4a presents the IR spectra of the pure components used for the preparation of the tablets. Further, in Figure 4b, the IR spectrum of CLZ and the IR spectra of newly prepared formulations have been compared.

All evaluated spectra present the specific bands of the main groups from the studied components, and their position and assignment are listed in Table 2.

As can be seen from Figure 4b, the IR spectra of formulations closely resemble the spectrum of CLZ attributable to the high content of this component in the tablets and the bands’ strong intensities. The influence of the other components is evidenced, especially in the spectral region 3000–3500 cm^−1^. Here, in the spectra of formulations, a large band with a maximum at 3467 cm^−1^ and a shoulder at 3413 cm^−1^ was observed.

The first one represents a superposition of the OH groups from CLZ, KOL, and CHT, while the second one belongs to OH groups from LIG. At the same time, the bands located at 3204 and 3156 cm^−1^ assigned to stretching vibration of NH groups present a slow increase in width due to the superposition of these bands with the OH bands of KOL, CHT, and LIG. In the 3000–2700 cm^−1^ region of the IR spectra of formulations, two bands at 2923 and 2857 cm^−1^ can be observed. These can arise from the superposition of the symmetric and asymmetric stretching vibrations of CH groups from LIG, CHT, and KOL over those of CLZ.

In the fingerprint region of the IR spectra of the formulations, the bands from 1358 and 1471 cm^−1^, which are assigned to the stretching vibration of CC and CN groups, deformation vibration of CCH groups, and stretching vibration of CC groups, and deformation vibration of CCH and CNH groups from CLZ, are slowly shifted to higher wavenumber values. Two other small bands can be evidenced at 1394 cm^−1^, belonging to the superposition of the stretching vibration of CC groups and the deformation vibration of CCH and CNH groups from KOL with the stretching and deformation vibrations of CH groups from CHT and AV, and at 1023 cm^−1^, which arises from the superposition of Alkyl–O ether vibrations methoxyl and β–O–4 in guaiacol from LIG and the stretching vibration of CC and CCl groups and the deformation vibration of CCH groups from KOL. The second band increases slowly with the increase in the LIG content in the formulations.

The IR spectra of the formulations (Figure 4b) compared to pure components (Figure 4a) did not evidence further modifications, indicating that there were no visible interactions between the components of the tablets.

#### 2.2.2. X-Ray Diffraction

Figure 5a and Figure 5b present the X-ray diffractograms of the pure components and the newly prepared formulations, respectively.

The diffractogram of CLZ presents intense signals at 12.98°, 13.86°, 15.57°, 17.83°, 19.93°, 21.05°, 25.22°, 25.79°, 27.51°, and 32.07° (2θ), while KOL and CHT have a background pattern with two extense bands at 13.04° and 21.95° (2θ) and 9.32° and 20.01° (2θ), respectively. The LIG presents an amorphous background, being a tridimensional crosslinked polymer that usually does not present any crystalline phase; ST shows four intense signals at 1.79, 3.58, 5.36, and 21.46 (2θ degrees), and the AV diffractogram presents the specific signals assigned to cellulose at 15.1°, 16.4°, and 22.7° (2θ degrees).

The obtained formulations present characteristic signals of the pure components with no modification in the values of the 2θ degrees, indicating again (as in the case of infrared spectroscopy) that no chemical interactions occurred between the drug and other components during mixing and the tablet preparation process (Figure 5b).

The signals of CLZ decreased slightly in intensity in formulations due to a lower drug concentration (50% CLZ in the formulations compared to pure component).

The obtained formulations exhibit characteristic signals of the pure components without alterations in the 2θ degree values, indicating that no chemical interactions occur between the drug and the other constituents.

Furthermore, the crystallinity degrees (%) of the pure components and the formulations were calculated and are presented in Table 3.

As can be observed, the crystallinity degree of the formulations was influenced by the decrease in the concentration of the semicrystalline compound (AV) and the increase in the concentration of the amorphous compound (LIG).

#### 2.2.3. Thermal Characterization

##### TG/DTG/DTA Characterization

Figure 6a,b present a comparative representation of the TG curves for LIG, CLZ, CHT, AV, KOL, ST, and the formulations (Fla, FLb, and FLc). Based on the recorded TG, DTG, and DTA curves and their interpretation using the STAR software 19 from Mettler Toledo, the main thermogravimetric characteristics were obtained, as presented in Table 4.

The thermal behavior of CLZ, CHT, AV, KOL, and ST were presented in a previous study [29]. In this study, we additionally used LIG, whose thermal decomposition occurs through a series of six stages. The initial stages are associated with the removal of moisture and other volatile compounds.

The thermal decomposition of the LIG sample begins at 212 °C and proceeds through a series of three stages up to 533 °C. The most notable peak is observed at 350 °C, similar to those reported in other studies in the literature [38,39].

As shown in Figure 6a, lignin degradation occurs slowly, resulting in a residue of 60% at the end of the test, a value comparable to other results in the literature [40]. A large amount of obtained residue suggests a high cross-linking capacity and the possibility of polymerization at high temperatures [39,41].

The thermal decomposition of the FLa, FLb, and FLc formulations occurred in four stages. The second stage, with a peak temperature of approximately 284 °C, could be associated with the decomposition of CLZ in the temperature range of 227–306 °C. Analyzing the results presented in Table 5 and the DTG and TG curves in Figure 6b and Figure 7, we observed that the FLa, FLb, and FLc formulations exhibited roughly the same behavior up to approximately 410 °C, indicating a similar composition.

The temperature range of 305–362 °C is characteristic of the thermal decomposition of the used excipients (AV, CHT, LIG, and KOL). In contrast, the final decomposition stage was specific to the thermal decomposition of the excipient KOL.

According to the thermogravimetric curves presented comparatively in Figure 7, more significant differences were observed between the curves for FLa, FLb, and FLc in the final decomposition stage. Additionally, in the case of FLa, the T_peak_ temperature appears approximately four degrees lower than that for FLb and FLc, and the percentage mass loss is more significant than in the other two formulations.

##### DSC Characterization

DSC curves were recorded in a nitrogen atmosphere with a heating rate of 10 °C/min, including two heating cycles and one cooling cycle. The results obtained for LIG during the first heating cycle are presented in Figure 8. Two endothermic peaks are highlighted, consistent with the previously presented results for the thermogravimetric analysis. No other thermal effects appeared on the cooling curve or the second heating curve for the LIG sample. The first endothermic peak is characteristic of the moisture removal from the sample. At the same time, the second peak could be associated with the desorption of chemically bound water or the removal of other volatile compounds [39].

Figure 9a–c present the DSC curves comparatively obtained for the FLa, FLb, and FLc formulations.

The DSC curves for the first heating, shown in Figure 7a, confirmed the presence of the active substance, CLZ. A reduction in the melting peak temperature of CLZ is observed, from 191 °C to 170 °C, along with a decrease in the intensity of the process, indicating the presence of interactions between CLZ and the excipients [29].

The peak at 144, 145 °C, indicating the presence of LIG, was also notable. The variations in enthalpy values for these processes are −0.80 J/g for FLa, −1.29 J/g for FLb, and −1.61 J/g for FLc. In the case of the FLc formulation, the peak was broader, suggesting a higher amount of LIG incorporated into this sample. The peak at 44 °C was specific to the KOL excipient, while the peak at 70, 79 °C could be associated with moisture removal.

The analysis of the DSC cooling curves shown in Figure 7b indicated the presence of CLZ crystallization peaks, but at lower temperatures compared to CLZ in the absence of excipients (181 °C) [29], specifically at 129 °C for FLa, 122 °C for FLb, and 123 °C for FLc. Variations of enthalpy values were observed for the crystallization process associated with these peaks: 13.63 J/g for FLa, 8.10 J/g for FLb, and 2.40 J/g for FLc. For the FLc sample, a different behavior was noted, namely the fractional crystallization of CLZ in two steps, with one peak at 106 °C and another at 123 °C. The higher LIG content in this formulation likely contributed to the formation of a metastable polymorphic phase and the fractional crystallization of CLZ.

Figure 7c presents the characteristic curves of the second heating stage. The melting peaks at 158 °C, 152 °C, and 151 °C are specific CLZ, while the glass transition temperatures at 32 °C and 114 °C can be associated with the presence of KOL [42].

### 2.3. In Vitro Dissolution Studies

The results obtained during the in vitro dissolution test indicated the prolonged release of CLZ from the studied formulations compared to the release of CLZ from an industrial product [43] and the previously developed tablets containing a matrix of KOL and CHT. These results also highlighted the central role of the LIG and KOL exerted on matrix tablet release characteristics. The released amount of CLZ varied with the percentage of LIG and KOL for all studied formulations (Figure 10).

A particular behavior was observed for FLc containing 13 (*w*/*w*) % LIG, 30 (*w*/*w*) % KOL, and 5 (*w*/*w*) % CHT, as it released 28.42 (*w*/*w*) % of CLZ during the first two hours of the dissolution test in simulated gastric fluid. Moreover, the formulation released 94.50 (*w*/*w*) % CLZ of the 12 h and 99.30 (*w*/*w*) % at the end of the 36 h of testing. Consequently, in the FLa and FLb formulations containing 11–12 (*w*/*w*) % LIG, 30 (*w*/*w*) % KOL, and 5 (*w*/*w*) % CHT, at the end of the dissolution test, the amount of CLZ was the lowest.

The FLa released 96.17 (*w*/*w*) % of CLZ, and the FLb released 98.75 (*w*/*w*) %, the lower concentration of LIG determining these values. The results are illustrated in Figure 10.

We compared the dissolution profiles of CLZ from the three formulations FLa, FLb, and FLc (containing LIG, KOL, and CHT in the matrix) with the CLZ dissolution profile from F2b in our previous work [29], which contains only KOL and CHT in the matrix, (Figure 11). This aspect highlights the role of LIG in the more efficient release of CLZ.

The dissolution profiles were compared, considering CLZ content from the studied formulations, by calculating the similarity factor f2 and the difference factor f1. Formulations FLa, FLb, and FLc were considered reference formulations, and formulations F1a,b-F2a, and F2b [29] in our previous work were considered test formulations.

The obtained results (Table 5) confirmed that LIG, along with CHT and KOL, influenced the release characteristics of the matrix tablets, and virtually each studied formulation had its particular release kinetics.

Although the similarities between the three release profiles of CLZ were confirmed by the value of the similarity factor (f2 = 57.9562, f2 = 58.0386), the difference factor had a value greater than 15 (f1 = 33.7543, f1 = 36.0455), which corresponded to a difference of more than 10% between the FLa, FLb, and FLc analyzed profiles.

### 2.4. Drug Release Kinetics

The CLZ release study was performed in two steps, modifying the pH of release media to cover the physiological pH range attributed from 1.2 pH buffer simulating the gastric fluid to 6.8 pH buffer mimicking the intestinal fluid (Figure 12). A gradual release process that extended over several hours (up to 36 h) was recorded, which could lead to slow CLZ diffusion through the prepared formulations in both pH buffer solutions. In the 12–36 h interval, the release profiles were close to reaching the equilibrium.

At the end of the release process, the total released amount (Qmax, %) of CLZ from the formulations ranged between 96.17% and 99.30% (Table 6), where the FLc formulation registered the highest drug released percentage.

The results of the in vitro release tests have been evaluated from the kinetic perspective (fitted with the Korsmeyer–Peppas equation), and the kinetic parameters (n and k) were calculated to establish the mechanism and rate of drug release of CLZ (Table 7). The equation was applied to the first (0–2 h) and second (2.5 h—36 h) steps of drug release.

After evaluating the initial phase of the kinetic step release profile, the value of release exponent n for all the formulations was found to be under 0.5 (n = 0.414–0.466), which was associated with a pseudo-Fickian mechanism. A different mechanism of the drug release, namely an anomalous or non-Fickian diffusion mechanism, was characteristic of the second step of the kinetic release profile, where the *n* value ranged between 0.5 and 1.

It can be observed that the values of release rate constant *k* were well correlated with the release profiles presented in Figure 12, where the highest values of 0.733 h^−n^ in the first step and of 0.187 h^−n^ in the second step were registered for the FLc sample, data that were suggestive for a faster drug release. In Table 7, the values of R^2^ range between 0.991 and 0.995. For R^2^, the value should be as close as possible to 1 to demonstrate a yield that is as good as possible for a formulation.

In a study by Raval and colleagues [44], the solubility of pure CLZ drug was higher in pH 6.8 buffer solutions, whereas it was low in pH 1.2 buffer solutions. However, in our study, the release rate constant, k, was different for both steps of the release profiles, remaining dependent on the pH of the buffer solution, meaning that after the incorporation of CLZ in the three formulations, the release by diffusion was influenced by the pH sensibility of the drug solubility. It can be concluded that the CLZ-loaded formulations are suitable for use as promising oral delivery systems that provide a controlled and prolonged release over 36 h.

### 2.5. Antioxidant Activity

The ability of the studied materials to scavenge free radicals was used to evaluate their antioxidant activity. This is usually mainly due to the phenolic hydroxyls present in the lignin structure, which can neutralize free radicals by electron or proton transfer [45]. Our results revealed 41.82% ± 1.06n hibition occurring because of the LIG presence.

Depending on the concentration of LIG, several publications [46] found an improvement over radical scavenging. As the LIG concentration rose, its ability to scavenge free radicals increased from 17.18% ± 1.05 to 19.54 ± 1.10 or 21.36% ± 1.06 (Figure 13). The increase was attributable to the presence of free phenolic hydroxyl groups, and the variety of functional groups from the LIG structure have influenced the scavenger activity [47,48].

LIG can be highly beneficial when incorporated into dietary supplement tablets due to its antioxidant properties or to add value as a pharmaceutical excipient [49].

### 2.6. Antimicrobial Activity

Phenolic compounds from LIG have the ability to disrupt the bacterial cell membrane upon contact [50,51]. Bacteria perish as a result of cell membrane rupture and subsequent release of cell contents, hence improving the bacteriostatic effect.

Figure 14 shows the results of antimicrobial susceptibility tests on Mueller–Hinton agar for three different concentrations of tested tablets (FLa, FLb, FLc).

The results demonstrated that susceptibility depends on the amount of lignin in the tablet matrix and the microbial species. All tablets exhibited higher activity at the highest concentration (600 μg/mL) against *Staphylococcus aureus*, *Escherichia coli*, and *Candida albicans*, with the most vigorous activity observed against *Candida albicans*. The tablets showed no effective activity against *Pseudomonas aeruginosa*. However, tablets with higher LIG content (FLc) demonstrated excellent antimicrobial activity overall, including against colistin-resistant *Klebsiella* spp., as shown in Figure 15. This suggests that LIG might offer a promising alternative for combating infections caused by multidrug-resistant bacteria.

At a concentration of 600 μg/mL, the tested compounds exhibited promising antimicrobial activity, although it was lower than that of the standard drugs Fluconazole (25 µg/mL) and Levofloxacin (5 μg/mL). Overall, the antimicrobial activity of the tablets increased with the amount of LIG in their matrices.

These results were attributed to the LIG content of the tablets tested, which seemed to be the main factor influencing the antibacterial activity. That property was indeed correlated to the ability of this compound to diffuse through biological membranes to reach its target of action.

The data presented showed that the amount of LIG in the matrix of tablets with CLZ increased the antimicrobial activity. The higher the concentration of LIG in the matrix, the higher the antimicrobial activity.

## 3. Materials and Methods

### 3.1. Materials

The following materials have been utilized: Chlorzoxazone (5-chloro-2,3-dihydro-1,3-benzoxazol-2-one) (Orchid Chemicals Ltd., Chennai, India), alkali lignin (Sigma Aldrich), Kollidon^®^ SR (BASF, Ludwigshafen, Germany), chitosan (practical grade, BASF, Germania), (Chemtrec, Falls Church, VA, USA), magnesium stearate (Union Derivan S.A., Barcelona, Spain) and 2,2′ diphenyl-1-picrylhydrazyl (DPPH) Sigma Aldrich; Dorset, UK.

Chlorzoxazone is a centrally acting muscle relaxant with sedative properties, and it is used for the symptomatic treatment of painful muscle spasms.

Kollidon^®^ SR is a physical mixture of 80% polyvinyl acetate (average molecular weight of 450,000 Daltons) and 20% polyvinyl pyrrolidone (povidone) (average molecular weight of 40,000 Daltons). Avicel ^®^PH-113 is a microcrystalline cellulose used for direct compression, and in wet, dry, and granulated states, it is a binder and compression aid.

### 3.2. Methods

#### Preparation of CLZ Matrix Tablets

Matrix tablets were prepared using the power mixtures for the three proposed formulations by direct compression with the Korsh EK0 compression machine (10 mm ponson diameter, 85–90 kN compression force) [52].

The compositions of the studied formulations are given in Table 1.

The powders corresponding to the formulation of matrix tablets with CLZ were mixed in an Erweka AR 403 mixer (Erweka GmbH, Heusenstamm, Germany) with a rotation speed of 400 rpm for 5 min. After that, they were sieved using an EM-8 electromagnetic sieve (Erweka GmbH, Heusenstamm, Germany). Then, they were subjected to compression directly to the Korsch EK0 compression machine (Korsch AG, Berlin, Germany).

Table 1 presents the compositions of the three oral formulations, where besides LIG, KOL, and CHT, other specific auxiliary excipients were included [53]. In 100 (*w*/*w*) % powder mixture 50 *w*/*w* % representing CLZ, the proportions of KOL 30 *w*/*w* %, CHT 5 (*w*/*w* %), and LIG 11–13 (*w*/*w* %) were varied, and the ST constant values and the AV were completed at 100 *w*/*w* %. These proportions were taken into account when preparing the tablets, but the average weight per tablet is 500 mg. A matrix tablet (250 mg) can contain 55 mg LIG (FLa), 60 mg LIG (FLb), or 65 mg LIG (FLc).

The matrix tablets with an average diameter of 10.0 mm and thickness of 3.0 mm have been obtained. The effects of LIG, KOL, and CHT were studied individually and in combination at 1:1 CLZ/excipient ratios. The influence of LIG on KOL and CHT and the yield of CLZ was studied. This strategy has evaluated the impact of therapeutic system type on the oral availability of CLZ and the antioxidant and antimicrobial actions due to lignin.

### 3.3. Evaluation of the Prepared Tablets

#### 3.3.1. Pharmaco-Technical Parameter of Matrix Tablets

The quality of the matrix tablets was assessed by determining the pharmaco-chemical characteristics of hydrophilic matrix tablets with the modified release [52,54]: weight uniformity, drug content uniformity, friability, mechanical strength, and thickness.

Weight Variation

Weight uniformity was determined according to the European Pharmacopoeia, 11th ed. [30] by weighing 20 tablets on the Radwag WPE 60 electronic balance.

Twenty tablets were randomly selected from each formulation and weighed individually. The individual weights were compared with the mean weight, and the standard deviation (SD) was evaluated [55].

Thickness, Diameter, and Mechanical Strength

The tablets’ pharmaco-technical characteristics (diameter, thickness, and average) were evaluated according to the European Pharmacopoeia, 11th ed. [30].

To determine their mass and dose uniformity, 29 tablets were weighed using an electronic balance, Redwag WPE 60.

Mechanical resistance was performed on ten tablets on a Schleuniger Pharmatron tablet hardness tester 8M (Sotax AG, Aesch, Switzerland).

From thickness variation, ten tablets from each formulation were taken randomly, and their thickness was measured using a micrometer (Micro-Epsilon Messtechnik GmbH & Co. KG, Ortenburg, Germany). The mean thickness and SD were calculated.

Friability Test

Friability was determined on 20 tablets on Schleuniger Pharmatron FTII friability tester (Sotax AG, Aesch, Switzerland) at 100 rpm for 4 min.

Hydration Capacity or Swelling Degree

These parameters were determined by using a dissolution test station type II, Hanson SR 8 Plus Series (Hanson Research Co., Chatsworth, CA, USA).

Matrix tablets were introduced in 1000 mL distilled water at 37 ± 2 °C at 60 rpm. At the predetermined time intervals (1 h), samples prevailed from the hydration medium, and the excess water on the surface was removed by wiping with filter paper, and then samples were weighed.

Dose Uniformity

The dose uniformity was evaluated by quantitatively determining CLZ in tablets using an HPLC method. It was detected by the UV spectrum at its characteristic wavelength of 280 nm. HPLC was performed using a chromatograph of Thermo Fisher Surveyor type (Thermo Fisher, San Jose, CA, USA) equipped with a UV-Vis detector with multiple diode array detectors and a Thermo Fisher-Hypersil Betasil C18 150 mm × 4.6 mm column (Thermo Fisher, San Jose, CA, USA), the particle size dimension was of 5 μm. The column temperature was kept constant at 45 ± 0.2 °C. As the mobile phase, a mixture of acetonitrile and water in the 20:80 *v*/*v* ratio was used at a flow rate of 1.0 mL min^−1^. The injection volume for each determination was 20 μL. The UV spectrum detected CLZ.

LIG Identification

LIG was determined by the HPLC method described above. Detection of LIG was performed at a wavelength of 216 nm.

#### 3.3.2. Drug-Excipients Compatibility Study

##### Fourier-Transform Infrared Spectroscopy

The IR spectra in the 4000–500 cm^−1^ spectral region with 4 cm^−1^ spectral resolution of pure components and the prepared formulations were recorded in transmittance mode, using KBr as the matrix, on a Bruker ALPHA FT-IR spectrometer. The pellets’ concentration was 3 mg sample/300 mg KBr for all studied formulations.

##### X-Ray Diffraction

The diffractometer D8 ADVANCE (Bruker AXS, Karlsruhe, Germany) was used to record the diffractograms. The working conditions were 40 kV, 30 mA, a 2 s/step, and 0.02 degrees/step. All diffractograms were recorded in the 5–50° 2θ range at room temperature.

##### Thermal Characterization

Using a Mettler Toledo 851e device (Columbus, OH, USA), thermogravimetric (TG), derivative thermogravimetric (DTG), and differential thermal (DTA) curves were recorded for formulations containing alkali lignin. These measurements were recorded within a temperature range of 25–700 °C in an inert atmosphere, with a heating rate of 10 °C/min. The sample mass varied between 4 and 4.6 mg, and the nitrogen flow rate was 20 mL/min.

Additionally, differential scanning calorimetry (DSC) curves were recorded with a Mettler Toledo DSC1 device. The same heating rate was used as in the thermogravimetric analysis, namely 10 °C/min in an inert atmosphere (nitrogen). Two heating cycles and one cooling cycle were performed. The scans were carried out over a temperature range of −20 to 180 °C for formulations (FLa and FLb), −20 to 200 °C for formulation FLc, and between 25 and 160 °C for the LIG sample. The sample mass varied between 4 and 5 mg.

The TG, DTG, DTA, and DSC curves were interpreted using the STAR software from Mettler Toledo to obtain the main thermal characteristics.

#### 3.3.3. In Vitro Dissolution Studies

The CLZ dissolution profiles from the matrix tablets have been studied in two dissolution media with different pH, namely HCl 0.1 N solution with pH = 1.2 (simulating media for gastric fluids) and phosphate buffer solution with pH = 6.8 (simulating media for intestinal fluids). The experiments were carried out by means of a dissolution test station type II, Hanson SR 8 Plus Series (Hanson Research Co., Chatsworth, CA, USA), provided with two blades. The experiments were performed according to the requirements described in European and United States Pharmacopoeia (USP, with specifications for liquid and solid pharmaceutical formulations) [30,54].

The dissolution medium was a pH 1.2 solution for the first two hours and a pH 6.8 solution (phosphate buffer) for the next 36 h. SR 8 Plus Series blades apparatus type II was set at 37 ± 0.5 °C and 50 rpm; the sampling interval was set every hour during the 36 h test (5 mL of sample was replaced with the same volume of medium). The quantitative determination of CLZ was performed using an HPLC method.

The results were interpreted and statistically analyzed using Matlab 7.9. According to pharmaco-technical specifications for the preparation of modified-release tablets, the release profiles of the active substance from such tablets must be examined by determining the dissolution test results for solid pharmaceutical forms, difference factor f1, similarity factor f2, and correlation coefficient R^2^ between two or more formulations [56,57,58].

The difference factor, f1, and the similarity factor, f2, [59] have been calculated according to the following equations (Equations (1) and (2)):(1)$$f_{1}=\left\{\frac{\sum_{t=1}^{n}\left|R_{t}-T_{t}\right|}{\sum_{t=1}^{n}R_{t}}\right\}\times 100$$
and(2)$$f_{2}=50\log_{10}\left\{{\left[1+\frac{1}{n}\sum_{t=1}^{n}{\left(R_{t}-T_{t}\right)}^{2}\right]}^{-0.5}\times 100\right\}$$
where n is the number of points for specimen collection, R_t_ is the amount of dissolved active substance from the reference formulation at the t moment, T_t_ = the amount of dissolved active substance of the studied formulation at the t moment, and log10X = 10 base logarithm of X.

#### 3.3.4. Drug Release Kinetics

The drug release data up to 60% of the total drug released were fitted using the Korsmeyer–Peppas equation (Equation (3)) to establish the release mechanism [60]:(3)$$\frac{M_{t}}{M_{\infty }}=kt^{n}$$
where *M_t_*/*M_∞_* = fraction of the drug released at time *t*, *M_t_* = absolute cumulative amount of drug released at time *t*, *M_∞_* = maximum amount released in the experimental conditions used at the plateau of the release curves, *k* = release constant, and *n* = release exponent, which is indicative of the release mechanism.

In the above equation, a value of n = 0.5 indicates a Fickian diffusion mechanism of the drug from the samples, while a value of 0.5 < n < 1 indicates anomalous or non-Fickian behavior. When n = 1, a case II transport mechanism is involved with zero order kinetics, while n > 1 indicates a unique case II transport mechanism [61].

To analyze the mechanism of drug release from matrix tablets, the results of in vitro release data were plotted using various kinetic models, such as zero-order, first-order, Higuchi, and Korsmeyer–Peppas models.

The evaluation of CLZ release profile kinetics from matrix tablets based on LIG, KOL, and CHT was conducted using analysis by fitting into four representative mathematical models, which are essential elements for understanding the mechanisms underlying the release of active substances from the studied formulations.

The data fitting was carried out by linear or non-linear regression using Matlab 7.1. The correlation coefficient R^2^ (Equation (4)) was the criteria for selecting the model that most faithfully depicted the release profile of each studied formulation. A prediction as good as possible for the model requires R^2^ to be as close to 1 [62].(4)R2=1−∑i−1nyi−y^;i2∑i=1nyi−y¯2
where y_i_ = experimental data, y^i = values approximated by model, and Ӯ = average of experimental data.

#### 3.3.5. Antioxidant Activity

Lignin contains many functional groups, including hydroxyl, methoxyl, carbonyl, carboxyl, and quinone groups. Based on these groups, lignin and its derivatives can be used to create materials with ultraviolet absorption, biodegradability, antibacterial, antioxidant, electron transfer, and metal adsorption properties [63].

DPPH (2,2-diphenyl-1-picrylhydrozyl) (Sigma Aldrich; Dorset, UK) radical was employed to measure the antioxidant activity of the produced tablets based on the radical scavenging property of the LIG [64].

The DPPH assay was carried out in accordance with the procedure described by Jan et al. [65]. The violet DPPH radical, stable in dimethyl sulfoxide: water solution, is transformed after its reduction in the presence of LIG to a pale-yellow product, whose absorbance is measured at 517 nm. The results expressed scavenging ability (%). Firstly, 100 mL of dimethyl sulfoxide (Sigma Aldrich; Dorset, UK)/water (10:90, *v*/*v*) was used to dissolve 2.5 mg of DPPH. LIG has dissolved in a dimethyl sulfoxide/water (10:90, *v*/*v*) solution with a concentration of 20 mg/mL. In regard to the effect of the CLZ on the antioxidant properties, a mixture containing combination at 1:1 CLZ/excipient ratios as well as CLZ in the same proportions as the produced tablets was also dissolved in a dimethyl sulfoxide/water (10:90, *v*/*v*) solution with a total concentration of 75 mg/mL. Then, 50 μL of each mixture was mixed with 150 μL of the DPPH solution and made up to 3 mL with dimethyl sulfoxide/water (10:90, *v*/*v*) solution. After 60 min of dark incubation, the mixture was tested for absorbance at 517 nm in triplicate using a UV–Vis (Thermo Fisher, San Jose, CA, USA). A control sample of 0.025 mg/mL of DPPH was also evaluated. The radical scavenging activity (DPPH inhibition) was calculated as Equation (5) [64].% inhibition = (A_control_ − A_sample_/A_control_) × 100(5)
where: A_control_ and A_sample_ were the absorption intensities for blank and sample probes.

#### 3.3.6. Antimicrobial Activity

The new tablets based on CLZ containing 11, 12, 13% (*w*/*w*) in their LIG matrices were evaluated for their antimicrobial activity against Gram-positive bacteria: *Staphylococcus aureus* (ATCC 25923), *Escherichia coli* (ATCC 25922), *Pseudomonas aeruginosa* (ATCC 27853), *Candida albicans* (ATCC 10311), and two clinical isolates of *Klebsiella* spp. that are resistant to colistin.

The compounds’ antimicrobial activity was qualitatively evaluated using the agar diffusion method, following the guidelines for antimicrobial susceptibility testing established by the Clinical and Laboratory Standards Institute (CLSI) [66]. We used the cylinder technique for testing, and the antimicrobial activity was estimated by measuring the diameter of the area inhabited by the tested new tablets.

Microbial suspensions were prepared in sterile peptone water from 24 h cultures and standardized to a turbidity of 0.5 McFarland (approximately 1 × 1081\times 10^81 × 108 CFU/mL). These suspensions were used to inoculate 90 mm Mueller–Hinton agar plates.

The tested compounds were dissolved in dimethyl sulfoxide (DMSO) to achieve a final concentration of 600 μg/mL. Wells of 6 mm diameter were aseptically bored into the agar, and 0.1 mL of the compound solutions were carefully introduced into each well using metal cylinders placed on the agar surface. The plates were incubated at 37 °C for 24 h, allowing the compounds to diffuse into the agar and interact with the inoculated microorganisms.

For comparison, standard commercial disks containing Fluconazole (25 µg/mL) and Levofloxacin (5 µg/mL) were used as positive controls. Each microorganism was tested in triplicate. After incubation at 37 °C for 24 h, the diameters of the inhibition zones around the wells were measured and reported as mean ± standard deviation (SD).

## 4. Conclusions

In the present work, we propose the use of LIG as an excipient for direct compression and the study of the influence of LIG on the modified/controlled release of CLZ from the hydrophilic matrices containing KOL and CHT. In the current study, three formulations of FLa, FLb, and FLc of matrix tablets with CLZ were obtained with modified release. The proportions that were considered when preparing the 500 mg tablets were matrix tablets (of 250 mg) capable of incorporating 55 mg LIG (FLa), 60 mg LIG (FLb), or 65 mg LIG (FLc).

LIG, KOL, and CHT as matrix-forming agents influenced the flow and the compressibility properties and they have shown a significant effect on the pharmaco-chemical characteristics of the matrix tablets. The pharmaco-chemical characteristics of the tablets, defined both by the working conditions and the flowing and compressibility characteristics, were directly influenced by the matrix-forming polymers used to obtain hydrophilic matrix tablets with modified release. The mechanical strength of the tablets varied directly to the LIG concentration in the formulation. The optimal concentration of excipients has been established, as that of KOL was 30 (*w*/*w*) %, and that of CHT was 5 (*w*/*w*) %. The effects of LIG, KOL, and CHT were studied individually, and they form the hydrophilic matrix that encapsulates CLZ in a 1:1 ratio. All three FLc formulations showed optimum pharmaco-technical properties, and they exhibited the most significant potential for use in oral pharmaceutical products for the controlled delivery of CLZ.

The FT-IR spectroscopy and XRD analysis revealed no interactions between active principle and excipients. KOL, CHT, LIG, ST, and AV used in this study did not interact with the CLZ and may be valuable excipients in pharmaceutical formulations. By increasing temperature, some interactions have been evidenced. The DSC curves for the first heating stage confirmed the presence of the active substance CLZ and its good compatibility with the used excipients. A reduction in the melting peak temperature of CLZ was observed, from 191 °C to 170 °C, along with a decrease in the intensity of the process, indicating the presence of interactions between CLZ and the excipients. The peak at 144, 145 °C, indicating the presence of LIG, was also notable. The higher LIG content in this formulation likely contributed to the formation of a metastable polymorphic phase and the fractional crystallization of CLZ.

The in vitro dissolution test proved that the CLZ tablets were suitable for the prolonged and controlled release of CLZ. The released amount of CLZ varied with the percentage of LIG for all studied formulations. The FLc containing 13 (*w*/*w*) % LIG, 30 (*w*/*w*) % KOL, and 5 (*w*/*w*) % CHT released 28.42 (*w*/*w*) % of CLZ during the first two hours of the dissolution test in simulated gastric fluid, 94.50 (*w*/*w*) % CLZ of the 12 h and 99.30 (*w*/*w*) % at the end of the 36 h of testing. The FLa formulation containing 11 (*w*/*w*) % LIG released 96.17 (*w*/*w*) % of CLZ and FLb formulations containing 12 (*w*/*w*) % LIG released 98.75 (*w*/*w*) % CLZ at the end of the dissolution test.

The values of the difference factor f1, the similarity factor f2, and the correlation coefficient R^2^ showed that the three formulations studied confirmed that LIG next to KOL and CHT influenced the release characteristics of the matrix tablets, and virtually each studied formulation had its particular release kinetics.

The in vitro kinetic study revealed a complex mechanism of release occurring in two steps. Evaluating the first step of the kinetic step release profile is associated with a pseudo-Fickian mechanism, and the non-Fickian diffusion mechanism was characteristic of the second step. The incorporated formulations with CLZ are suitable for use as oral delivery systems that provide a controlled and prolonged release over 36 h. In vitro, dissolution tests revealed that the values of release rate constant *k* are well correlated with the release profiles and values of R^2^, where the highest values were registered for the FLc sample. These data are suggestive of a faster drug release.

The ability of the studied hydrophilic matrices with LIG, KOL, and CHT to scavenge free radicals was used to evaluate their antioxidant activity. The three formulations FLa, FLb, and FLc of matrix tablets with CLZ developed good antioxidant activity.

The antioxidant activity of FLc was found to be 21.36% ± 1.06 greater than those of the FLa and FLb; the amount of LIG in the matrix of tablets with CLZ improved the antimicrobial activity. The tablets FLa, FLb, and FLc presented higher activity on *Staphylococcus aureus*, *Escherichia coli*, *Candida albicans*, and colistin-resistant *Klebsiella* spp. The most potent activity was noticed in relation to *Candida albicans*. The higher the concentration of LIG in the matrix (FLc), the higher the antimicrobial activity.

In conclusion, the results confirmed that CLZ can be formulated as hydrophilic matrix tablets based on LIG, KOL, and CHT. LIG can be highly beneficial when incorporated into dietary supplement tablets due to its antioxidant and antimicrobial properties or it can add value as a cost-effective pharmaceutical excipient owing to the reduced expense of lignin.

## Figures and Tables

**Figure 1 molecules-30-01426-f001:**
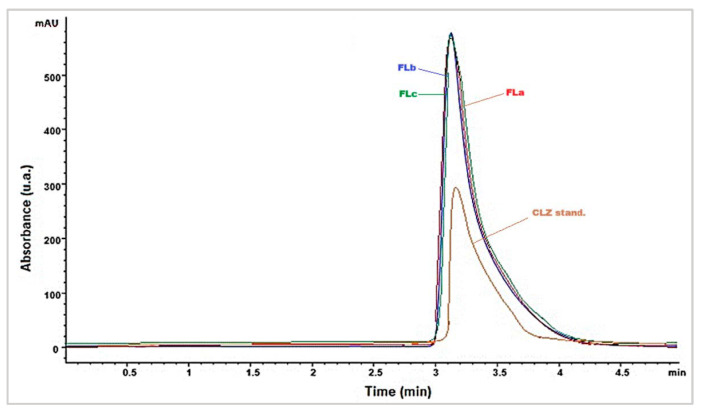
HPLC chromatograms of the CLZ and three formulations (Fla, FLb, FLc).

**Figure 2 molecules-30-01426-f002:**
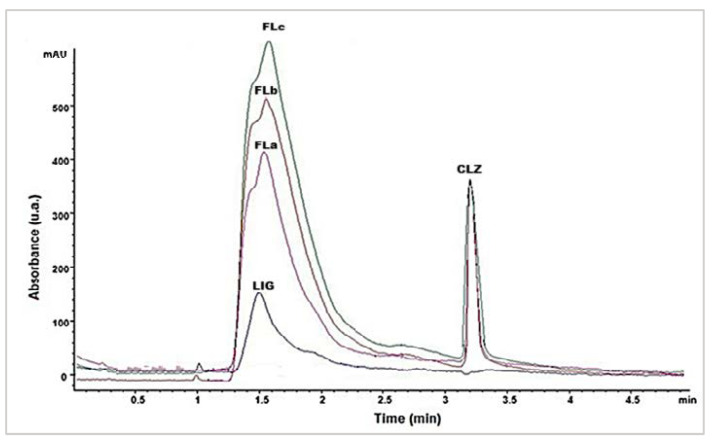
HPLC chromatograms of the LIG and CLZ and three formulations (Fla, FLb, FLc).

**Figure 3 molecules-30-01426-f003:**
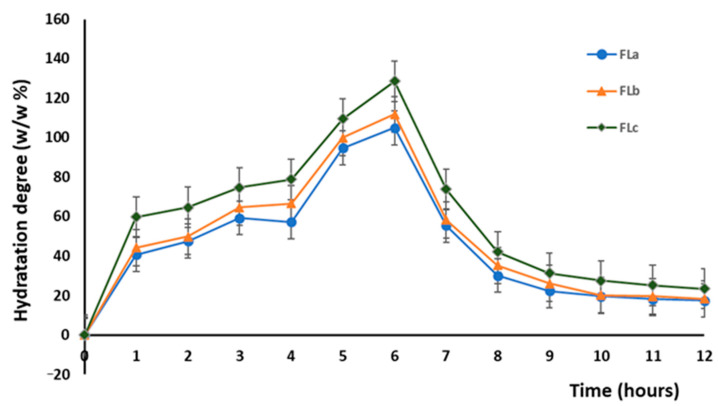
Variation in time of the hydration degree of the matrix tablets.

**Figure 4 molecules-30-01426-f004:**
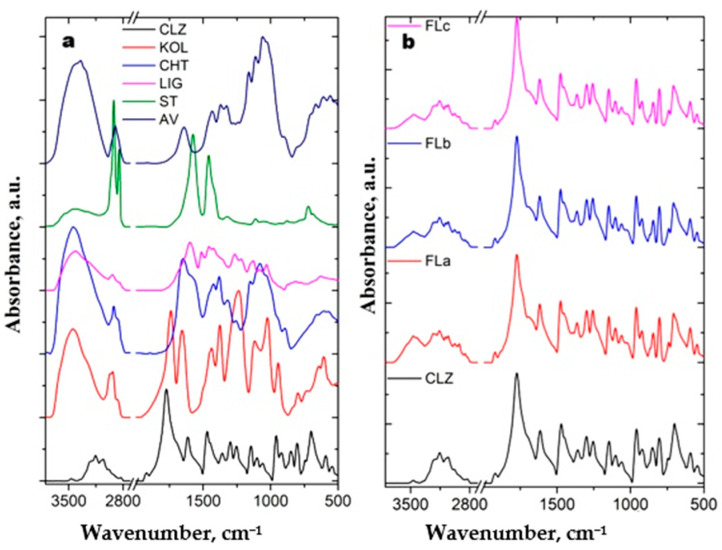
IR spectra of (**a**) pure components and (**b**) the comparison between CLZ spectrum with those of the formulations.

**Figure 5 molecules-30-01426-f005:**
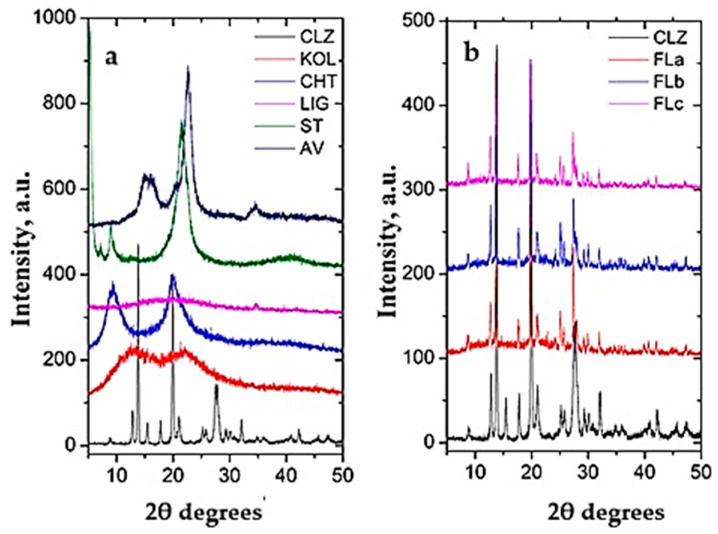
X-ray diffractograms of (**a**) pure components and (**b**) the comparison between CLZ and the formulations.

**Figure 6 molecules-30-01426-f006:**
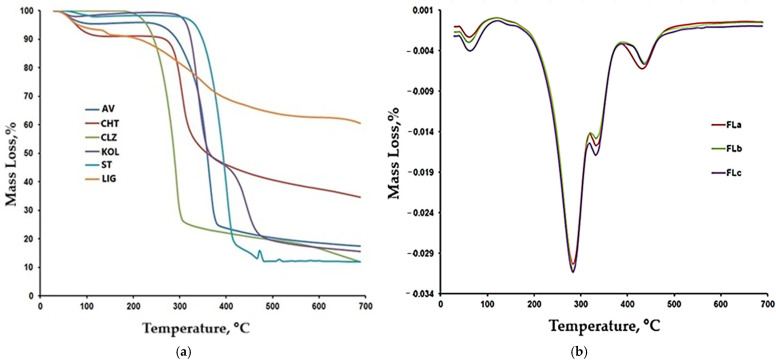
(**a**) TG curves for LIG, CLZ, CHT, AV, KOL, ST. (**b**) DTG curves for formulations (FLa, FLb, and FLc).

**Figure 7 molecules-30-01426-f007:**
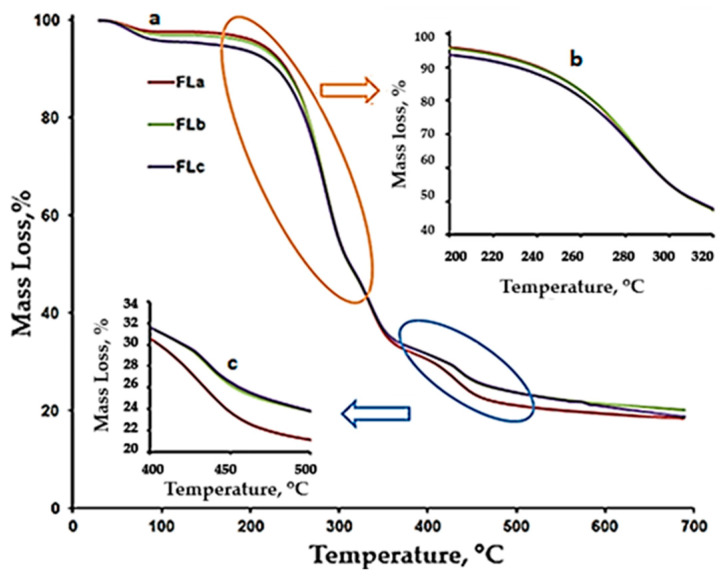
TG curves for the FLa, FLb, and FLc formulations: (**a**) (changes in the range 50–7000 °C), (**b**) (changes in the range 200–350 °C), (**c**) (changes in the range 400–700 °C).

**Figure 8 molecules-30-01426-f008:**
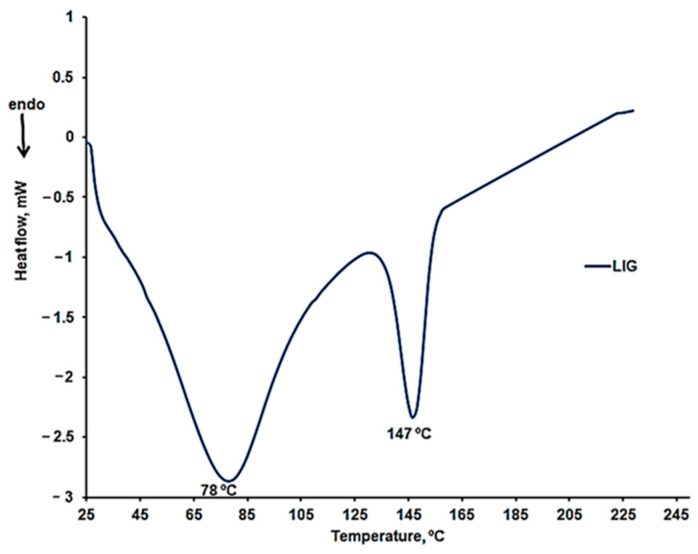
DSC curves for the first heating cycle of the LIG sample in the temperature range 25–160 °C.

**Figure 9 molecules-30-01426-f009:**
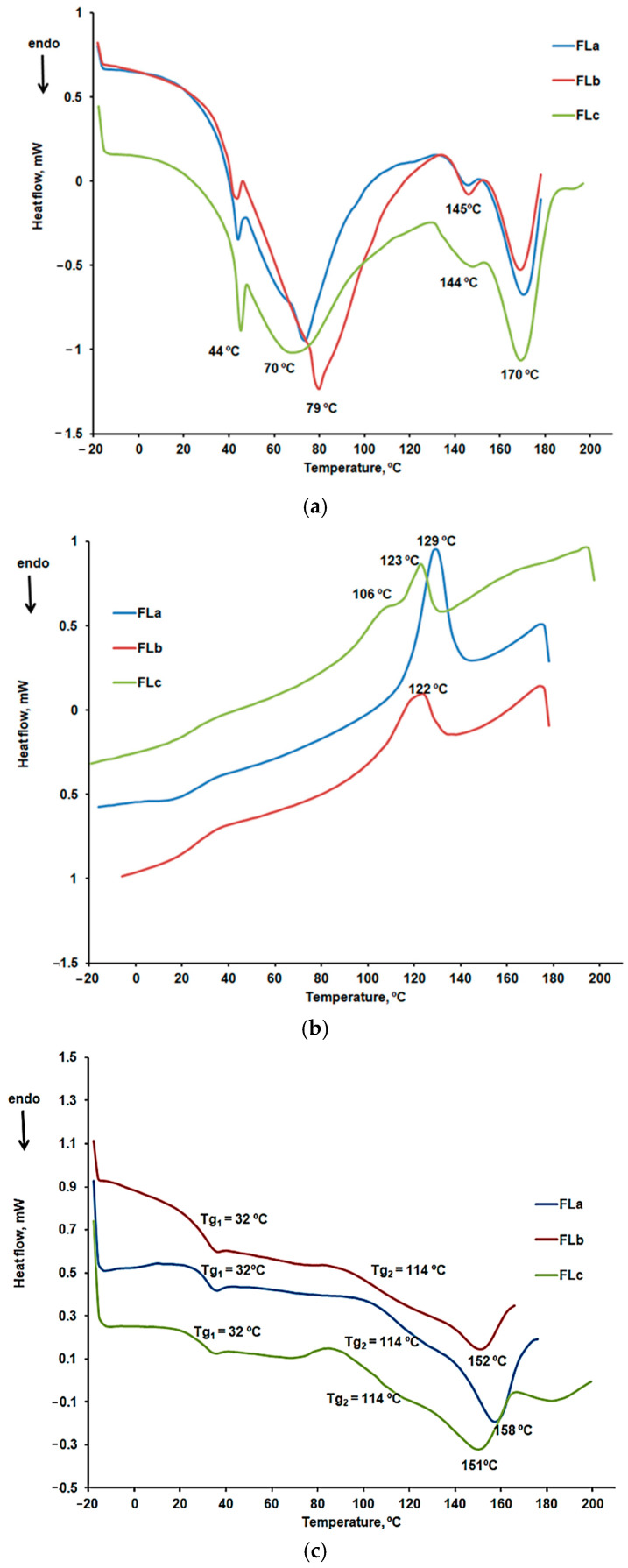
(**a**) The DSC curves obtained for the FLa, FLb, and FLc formulations in the first heating stage. (**b**) The DSC curves obtained for the FLa, FLb, and FLc formulations in the cooling stage. (**c**) The DSC curves obtained for the FLa, FLb, and FLc formulations in the second heating stage.

**Figure 10 molecules-30-01426-f010:**
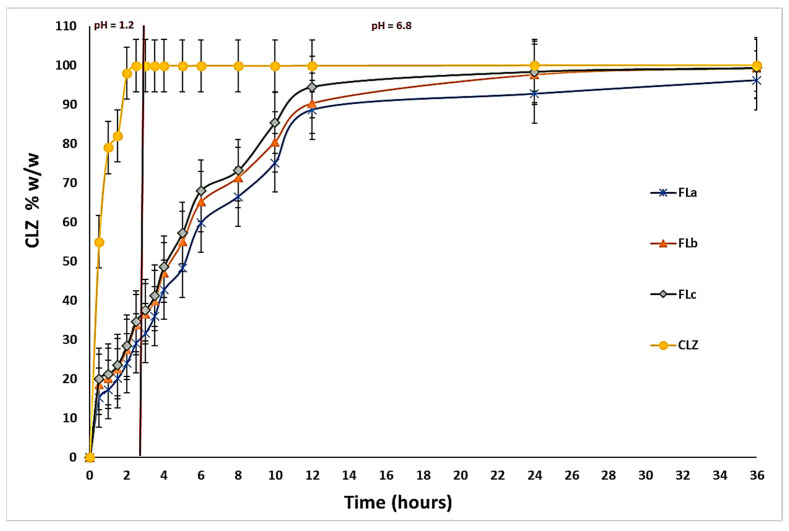
In vitro dissolution profiles of CLZ in FLa, FLb, and FLc compared to immediate-release CLZ tablets.

**Figure 11 molecules-30-01426-f011:**
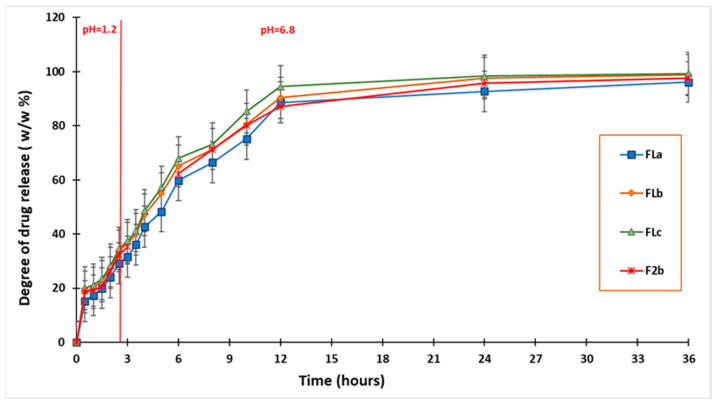
In vitro dissolution profiles of CLZ in Fla, FLb, and FLc compared to F2b.

**Figure 12 molecules-30-01426-f012:**
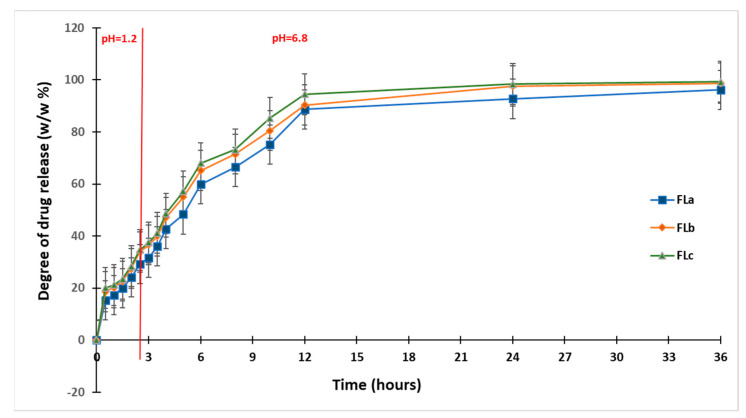
In vitro dissolution profiles of CLZ release mimicking the physiological path.

**Figure 13 molecules-30-01426-f013:**
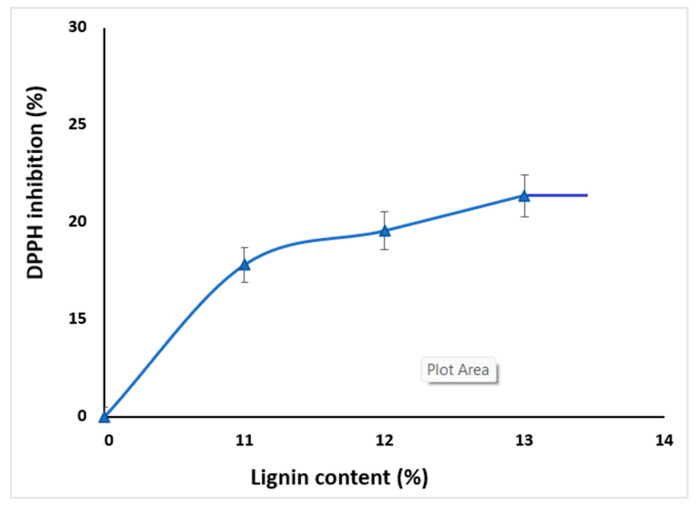
DPPH inhibition for LIG/KOL, CHT blends as a function of the LIG content.

**Figure 14 molecules-30-01426-f014:**
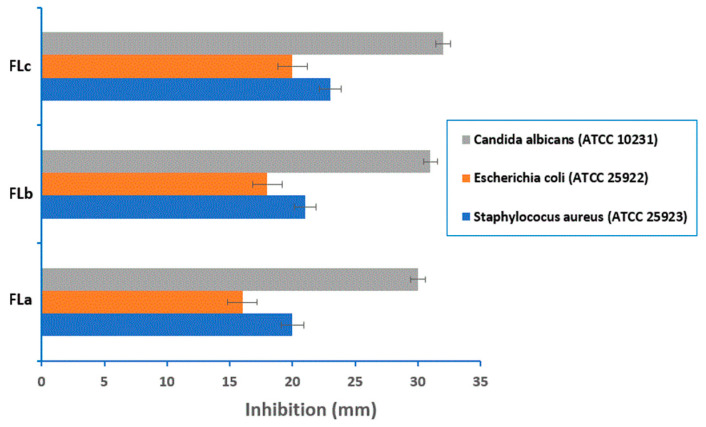
In vitro antimicrobial activity of the tested tablets FLa, FLb, FLc.

**Figure 15 molecules-30-01426-f015:**
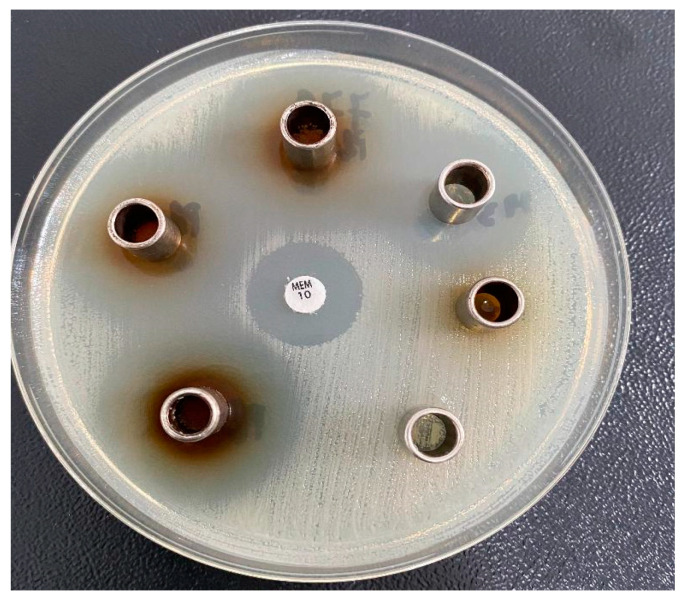
The action of LIG on a clinical isolate of *Klebsiella* spp. resistant to colistin.

**Table 1 molecules-30-01426-t001:** The composition and pharmaco-technical parameters of tablet formulations.

Formulations	Without Lignin	FLa	FLb	FLc
Composition mg/cp; (*w*/*w* %)				
CLZ	250 (50%)	250 (50%)	250 (50%)	250 (50%)
KOL	150 (30%)	150 (30%)	150 (30%)	150 (30%)
CHT	25 (5%)	25 (5%)	25 (5%)	25 (5%)
LIG	0	55 (11%)	60 (12%)	65 (13%)
ST	2.5 (0.5%)	2.5 (0.5%)	2.5 (0.5%)	2.5 (0.5%)
AV	72.5 (14.5%)	17.5(3.5%)	12.5 (2.5%)	7.5 (1.5%)
Parameters				
Diameter (mm)	10.133 ± 0.025	10.121 ± 0.022	10.061 ± 0.015	10.053 ± 0.012
Thickness (mm)	2.908 ± 0.173	2.914 ± 0.185	2.802 ± 0.112	2.773 ± 0.117
Average mass (g)	0.501 ± 0.089	0.493 ± 0.061	0.486 ± 0.084	0.488 ± 0.070
Dose uniformity (mg)	249 ± 0.932	248 ± 0.912	249 ± 0.963	250 ± 0.028
Mechanical strength (N)	86.83 ± 2.422	85.73 ± 2.532	87.55 ± 2.283	89.62 ± 2.311
Friability (%)	1.309 ± 0.041	1.312 ± 0.033	1.253 ± 0.027	0.891 ± 0.023

**Table 2 molecules-30-01426-t002:** The position of the IR bands and their assignments for all studied compounds.

Compound	IR Band Position	Band Assignment	Reference
CLZ	3468, 3204, and 3156 cm^−1^	stretching vibration of NH groups	[32]
	3080 and 3055 cm^−1^	stretching vibration of Ph-H groups	
	2982 and 2827 cm^−1^	stretching vibration of CH groups	
	1773 cm^−1^	stretching vibration of C=O groups	
	1614 cm^−1^	stretching vibration of C=C from aromatic rings	
	1471 cm^−1^	stretching vibration of C-C and CN groups and deformation vibration of CCH groups	
	1359 and 1255 cm^−1^	stretching vibration of CC groups and deformation vibration of CCH and CNH groups	
	1100 cm^−1^	stretching vibration of CN and CO groups and deformation vibration of CCH and CCC groups	
	1061 cm^−1^	stretching vibration of CC and CCl groups and deformation vibration of CCH groups	
	958 cm^−1^	stretching vibration of CN groups and deformation vibration of CCC groups	
	591 cm^−1^	deformation vibration of CNO, CCO, and CCN groups	
	545 cm^−1^	stretching vibration of CO and CCl groups and deformation vibration of CCC groups	
KOL	3447 cm^−1^	stretching vibration of OH groups	[33]
	2965, 2933, and 2870 cm^−1^	symmetric and asymmetric stretching vibration of CH groups	
	1740 and 1658 cm^−1^	stretching vibration of C=O groups from vinyl acetate and pyrrolidone ring	
	1375 cm^−1^	stretching vibration of COO groups	
	1236 cm^−1^	stretching vibration of CC groups and deformation vibration of CCH and CNH groups	
	1119 cm^−1^	stretching vibration of CN and CO groups and deformation vibration of CCH and CCC groups	
	1024 cm^−1^	stretching vibration of CC and CCl groups and deformation vibration of CCH groups	
	944 cm^−1^	stretching vibration of CN groups	
CHT	3444 cm^−1^	stretching vibrations of OH bonds	[34]
	2960, 2923, 2886, and 2866 cm^−1^	symmetric and asymmetric stretching vibrations of CH groups	
	1651 cm^−1^, 1597 cm^−1^, 1424 cm^−1^, and 1381 cm^−1^	stretching vibration of C=O groups of amide I, deformation vibrations of the NH (N-acetylated residues, amide II band), deformation vibration of CH_2_, CH_3_ groups and deformation vibration of OH groups	
	1258 cm^−1^	stretching vibration of NH, COC, and COH groups	
	1158 cm^−1^ and 1079 cm^−1^	stretching vibration of CO and COC groups	
LIG	3416 cm^−1^	stretching vibration of OH groups in phenolic and aliphatic structures	[35]
	2934 cm^−1^	stretching vibration of CH in aromatic methoxyl groups and aliphatic CH_2_ and CH_3_ groups	
	2846 cm^−1^	stretching vibration of CH in aromatic methoxyl groups and CH_2_ and CH_3_ groups	
	1599 cm^−1^	stretching vibration of C=C groups of the aromatic ring (S), and deformation vibration of CH groups	
	1511 cm^−1^	stretching vibration of C=C groups of the aromatic ring (G), CH deformation	
	1457 cm^−1^	asymmetric deformation C–H groups in CH_2_ and CH_3_	
	1422 cm^−1^	asymmetric deformation of C–H groups in –OCH_3_	
	1364 cm^−1^	symmetric bending of C–H from methoxyl group, O–H and C–O of phenol and tertiary alcohol	
	1266 cm^−1^	guaiacol ring breathing, C–O stretch in lignin, C–O linkage in guaiacyl aromatic methoxyl groups	
	1222 cm^−1^	syringyl ring breathing with C–O stretching	
	1129 cm^−1^	aromatic C–H in-plane deformation (typical for S units), secondary alcohols, C=O stretch	
	1078 cm^−1^	C–O deformation in secondary alcohols and aliphatic ethers	
	1032 cm^−1^	alkyl–O ether vibrations in methoxyl and β–O–4 in guaiacol	
	845 cm^−1^	CH out of plane vibrations in positions 2, 5, and 6 of guaiacyl units	
ST	3436 cm^−1^	stretching vibrations of the associated water molecules	[36]
	2921 and 2852 cm^−1^	stretching vibration of CH groups	
	1576 and 1456 cm^−1^	symmetric and asymmetric stretching vibration of (COO-) groups	
AV	3413, 3346, 1434, and 1324 cm^−1^	stretching and in-plane deformation vibration of OH groups	[37]
	2901, 1371, and 1276 cm^−1^	stretching and deformation vibration of CH groups	
	1643 cm^−1^	stretching vibration of absorbed OH and conjugated CO groups	
	1243, 1163, 1113, 1061, and 1026 cm^−1^	stretching vibration of CO groups	
	901 cm^−1^	stretching vibration of β-glucosidic linkage between the sugar units	

**Table 3 molecules-30-01426-t003:** The crystallinity degrees (%) for pure components and the newly prepared formulations.

Sample	CLZ	KOL	CHT	LIG	ST	AV	FLa	FLb	FLc
Degree of crystallinity %	80.0	24.3	37.2	0.0	49.5	47.7	50.9	49.5	48.1

**Table 4 molecules-30-01426-t004:** Main thermal characteristics of the formulations.

Sample	Stage/ DTA Characteristic	T_onset_, °C	T_peak_, °C	T_endset_, °C	ΔW, %	Residue, %
LIG	I/endo	49	62	93	6.68	60.08
	II/endo	137	142	149	2.10	
	III/exo	212	288	327	13.14	
	IV/exo	327	350	378	7.39	
	V/exo	378	448	533	8.22	
	VI/exo	658	-	-	2.39	
FLa	I/endo	53	62	91	2.36	18.47
	II/exo	227	284	306	50.15	
	III/exo	327	336	357	15.18	
	IV/exo	408	434	465	13.84	
FLb	I/endo	49	60	87	2.97	18.61
	II/exo	234	285	305	44.77	
	III/exo	305	334	362	20.40	
	IV/exo	425	437	481	12.02	
FLc	I/endo	47	64	96	4.55	19.84
	II/exo	242	284	304	42.80	
	III/exo	304	335	354	21.00	
	IV/exo	424	438	511	13.04	

**Table 5 molecules-30-01426-t005:** Values of f1 and f2 factors for the matrix tablet formulations.

Reference Formulations	Test Formulations	Difference Factor f1	Similarity Factor f2
FLa	F1a	47.7516	31.2235
	F2a	41.8503	38.3546
FLb	F1b	36.0455	58.0386
	F2b	33.7543	57.9562
FLc	F1c	51.6215	20.8511
	F2c	52.1065	19.2135

**Table 6 molecules-30-01426-t006:** Release parameters of CLZ from the three formulations.

Formulation Name	Q_max_ [%]	T_1/2_ [h]
FLa	96.17	4.96
FLb	98.32	4.25
FLc	99.30	4.15
CLZ	100	0.44

Q_max_ = maximum release amount; T_1/2_ = half release time.

**Table 7 molecules-30-01426-t007:** Kinetic parameters of CLZ release from investigated samples.

Sample Name	n	R^2^_n_	k [h^−n^]	R^2^_k_
First step of kinetic release profile (0–2 h)—pH 1.2				
FLa	0.466	0.989	0.713	0.999
FLb	0.444	0.962	0.716	0.997
FLc	0.414	0.967	0.733	0.998
CLZ	0.403	0.983	0.745	0.996
Second step of kinetic release profile (2.5 h–36 h)–pH 6.8				
FLa	0.684	0.990	0.162	0.996
FLb	0.645	0.991	0.184	0.994
FLc	0.656	0.991	0.187	0.995

n = release exponent, k = release rate constant, R^2^_n_ and R^2^_k_ = correlation coefficients corresponding to the slope obtained for determination of n and k.

## Data Availability

Available upon reasonable demand.

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
