# Peer review of "Lignin as a Bioactive Additive in Chlorzoxazone-Loaded Pharmaceutical Tablets"

_molecules, 2025, doi:10.3390/molecules30071426_

Round 1

Reviewer 1 Report

Comments and Suggestions for Authors

The authors have submitted a detailed article, which mainly focus on the influence of Lignin as an excipient for direct compression in the preparation of Chlorzoxazone-containing tablets. The data is rich, and the topic is meaningful. However, the author should check it more carefully as there are a lot of mistakes!

  1. Abstract, LIG, KOL and CHT, please provide full name when first appear.

Line 36, R2.

The microbial names should be Latin format (Line 41).

  1. How about the cost of LIG? Is it high priced as introduced in Line 62?
  2. In my opinion, Table 8 can be merged with Table 1, which make it easy to read. Moreover, why choosing 11-13% content of LIG? What is the basis for the selection?
  3. Line 293, it’s ‘Table 4’.
  4. Did you synthesize and test the parameters of tablet without LIG? For Figure 13, the point at origin (0 LIG content), is it the tablet without LIG? If so, please provide the main parameters in Table 1 as well.

     Overall,please check the manuscript carefully as there are a lot of mistakes that I have not list.

Author Response

Comment 1. Abstract, LIG, KOL and CHT, please providefull name when first appear.Line 36, R2.

Response 1: We proposed a shorter title for the article: Ligninas, a Bioactive Additive in Chlorzoxazone-Loaded Pharmaceutical Tablets. We have inserted the full names of the abbreviations when they appear.

Comment2: The microbial names should be Latin format(Line 41).

Response 2: Corrected by italics.

Comment 3: How about the cost of LIG? Is it high pricedas introduced in Line 62?

Response 3: Lignin has been deleted, and its lower cost has been mentioned in the conclusion.

Comment 4: In my opinion, Table 8 can be merged withTable 1, which make it easy to read. Moreover,why choosing 11-13% content of LIG? What isthe basis for the selection?

Response 4: The two tables have been merged as Table 1.Selection of the formulations detailed studied has be made taking in consideration thecompression processing parameters, which arekept at the values commonly employed.

Comment 5: Line 293, it’s ‘Table 4’.

Response 5: The correction has been made

Comment 6: Did you synthesize and test the parameters oftablet without LIG? For Figure 13, the point atorigin (0 LIG content), is it the tablet withoutLIG? If so, please provide the main parameters inTable 1 as well.

Response 6: A new column has been added in Table 1 with

the tested parameters of a tablet without LIG.

The authors wish to express their gratitude to the reviewer for the time dedicated to

evaluating this manuscript. We appreciate the feedback provided, which has significantly

contributed to the enhancement of the manuscript.

Reviewer 2 Report

Comments and Suggestions for Authors

This paper describes the effect of lignin on the release of chlorzoxane from a prepared pharmaceutical formulation. The work is quite extensive, the authors used many analytical techniques to characterize the obtained systems. Two main things need to be improved in my opinion. Although there are many drawings, the way they are prepared is not standardized. The axes are described in different fonts, with different sizes of letters, the units given in some drawings are barely visible (e.g. Fig. 2), also the legends in some drawings seem to be taken directly from the apparatus software (e.g. Fig. 2). My second point concerns literature, only 13 out of 61 references come from the last 5 years. The literature definitely needs to be refreshed to prove that the topic is current and interesting. 

Author Response

Reviewer 2

Comment 1: Although there are many drawings, the way they are prepared is not standardized. The axes are described in different fonts, with varying sizes of letters, the units given in some drawings are barely visible (e.g. Fig. 2), also the legends insome drawings seem to be taken directly fromthe apparatus software (e.g. Fig. 2).

Response 1: All drawings have been standardized, edited, and corrected. Thanks for your constructive comments.

Comment 2: My second point concerns literature, only 13 out of 61 references come from the last 5 years. The literature definitely needs to be refreshed to prove that the topic is current and interesting.

Response 2: Several recent references on this topic have been cited.

The authors wish to express their gratitude to the reviewer for the time dedicated to

evaluating this manuscript. We appreciate the feedback provided, which has significantly

contributed to the enhancement of the manuscript.

Reviewer 3 Report

Comments and Suggestions for Authors

The iThenticate report indicates that there is 50% repetition of wording in the manuscript. Authors must revise the manuscript according to the iThenticate protocol. Only after reducing the number of repetitions of phrases in the manuscript will it be possible to evaluate the manuscript.

Below are some considerations that must be taken into account:
1) From how many measurements are the results included in Table 1?
2) Figure 2- Describe the peaks in the HPLC chromatogram
3) Figure 4 - authors must indicate which of the figures is 4a and which is 4b

4) Figure 5 - authors must indicate which of the figures is 5a and which is 5b
5) From how many measurements are the results presented in Tables 3,4,5,6,7?
6) The measurement error should also be included in Figure 13

The research results must be discussed in detail based on the available scientific literature.

Author Response

Comment 1: The iThenticate report indicates that there is50% repetition of wording in the manuscript.Authors must revise the manuscript according tothe iThenticate protocol. Only after reducing thenumber of repetitions of phrases in themanuscript will it be possible to evaluate themanuscript.

Response 1:The text was corrected according to the iThenticate protocol.

Comment 2:The research results must be discussed in detailbased on the available scientific literature.The work submitted by the authors, entitled;The influence of Lignin in Chlorzoxazone Pharmaceutical Formulations obtained by direct compression & ", aims to create a modified release system for Chlorzoxazone by integrating Lignin into the formulation. This formulation appears to be interesting as it can lead to reduce the number of administrations of the active ingredient in a day. However, there are some critical points to be considered, as shown in the following observations, based on which an evaluation of the presented scientific work is provided.

Response 2: The title and some paragraphs in text have been modified.

The authors wish to express their gratitude to the reviewer for the time dedicated to

evaluating this manuscript. We appreciate the feedback provided, which has significantly

contributed to the enhancement of the manuscript.

Reviewer 4 Report

Comments and Suggestions for Authors

The work submitted by the authors, entitled "The influence of Lignin in Chlorzoxazone Pharmaceutical Formulations obtained by direct compression ", aims to create a modified release system for Chlorzoxazone by integrating Lignin into the formulation. This formulation appears to be interesting as it can lead to reduce the number of administrations of the active ingredient in a day. However, there are some critical points to be considered, as appear from the following observations, based on which is provided an evaluation of the presented scientific work.

Abstract

Line 40: Please correct the number of significant figures.

Introduction

Line 81-84: The self-citations inserted are related to the increase in solubility and stability of other drugs not to Chlorzoxazone.

Line 119: You say that Kollidon®SR is crospovidone but also a physical mixture of polymers. Please specify the composition.

Line 134: Please specify which Avicel®PH you used.

Results

Table 1: Except for friability, the three formulations are almost identical considering the SD.

Table 1: Please correct the number of significant figures.

Line 158 and 161: Why do you use the European Pharmacopoeia 8th edition and not the most recent one?

Figure 2: What do you mean by “HPLC chromatograms of the LIG and CLZ”? Did you prepare a sample containing both LIG and CLZ?

Line 198: Did you want to quote Table 1, maybe?

Figure 3: Why do you refer to the formulations F1a, F1b, and F1c and not FLa, FLb, and FLc? Also, the error bars are missing from the graph.

Line 207-208: The matrix tablets were almost unchanged in size and mass from the 10th hour. Please correct it.

Line 208-211: What do you mean when you say “depending on the content of the KOL and CHT”? In table 8 you report that the amount of KOL and CHT is the same in every formulation. Please rephrase the sentence to make it easier to understand.

Line 213-214: Please move this sentence to the paragraphs where you talk about the antioxidant and antimicrobial properties of the tablets.

Figure 4: The IR spectra of the formulations are almost superimposable with those of the drug. Also, the letters a and b are missing from the graphs.

Table 2: Please include this table as supplementary material.

Figure 5: The X-ray diffractograms of CLZ and the formulations are almost superimposable. Also, the letters a and b are missing from the graphs.

Line 284-286: How did you evaluate the degree of crystallinity of CLZ in the formulations?

Table 4: Please add to the caption that the main thermal characteristics were also evaluated for LIG.

Line 353, 362 and 371: Please correct the reference to the figures.

Figure 11: The error bars are missing from the graph. Also, this graph looks like the one in figure 12.

Figure 13: The error bars are missing from the graph.

Figure 14: Why is there no control?

Materials and Methods

Line 537: Wrong table is referenced, please correct it.

Line 576: Why do you write "hydration capacity or swelling degree" in the materials and methods paragraph but not in the results paragraph?

Line 703-704: Why do you say 1:1 ratio but you name LIG, KOL and CHT. Who is in 1:1 ratio? Please rephrase the sentence to make it easier to understand.

Line 715: What did you use as blank?

Line 719: Please correct “m/m” with “w/w”.

Conclusions

Line 758-759: What do you mean? Please explain it.

References

Correct the references in order to uniform them.

Throughout the text

A moderate revision of the language and typos is required. Also, images quality needs improving.

Comments on the Quality of English Language

The English could be improved to more clearly express the research.

Author Response

Commet:1Abstract Line 40: Please correct the number of significant figures.

Response:1Corrections done.

Commet:2Line 81-84: The self-citations inserted are relatedto the increase in solubility and stability of otherdrugs not to Chlorzoxazone

Response:2Aiswarya A., Suja C: Formulation and Evaluationof Chlorzoxazone Microspheres, EJPMR, 2021, 8(12), 379-389.

Commet:3 Line 119: You say that Kollidon®SR is crospovidone but also a physical mixture of polymers. Please specify the composition.

Response:3This information is added on page:  Kollidon®SR is a physical mixture of 80% polyvinyl acetate (average molecular weight of 450000 Daltons) and 20% polyvinyl pyrrolidone (povidone) (average molecular weight of 40000 Daltons).

Commet:4Line 134: Please specify which Avicel®PH you used.ResultsTable 1: Except for friability, the three formulations are almost identical considering the SD.

Response:4Avicel ®PH- 113 is a microcrystalline cellulose for direct compression and also in wet and dry state granulated, is a binder and compression aid.This was our task to keep properties of the tables in the same limits.

Commet:5Table 1: Please correct the number of significant figures.

Response:5European Pharmacopoeia 11th edition, 2023,278-280; 748-750.

Commet:6 Line 158 and 161: Why do you use the European Pharmacopoeia 8th edition and not the most recent one?

Response:6 European Pharmacopoeia 11th edition, 2023,278-280; 748-750.

Commet:7Figure 2: What do you mean by “HPLC chromatograms of the LIG and CLZ”? Did you prepare a sample containing both LIG and CLZ?

Response:7This has been recorded on a single component solution to assess the peaks.

Commet:8Line 198: Did you want to quote Table 1, maybe?

Response:8Corrected.

Commet:9Line 198: Did you want to quote Table 1, maybe?

Response:9Corrected.

Commet:10Figure 3: Why do you refer to the formulations F1a, F1b, and F1c and not FLa, FLb, and FLc? Also, the error bars are missing from the graph.

Response:10Corrected.

Commet:11:Line 207-208: The matrix tablets were almost unchanged in size and mass from the 10th hour. Please correct it.

Response:11:Corrected.

Comment12:Line 208-211: What do you mean when you say “depending on the content of the KOL and CHT”?

Response12:This has been deleted.

Comment13:In table 8 you report that the amount of KOL and CHT is the same in every formulation. Please rephrase the sentence to make it easier to understand.

Response13:Table 8 is deleted.

Comment14:Line 213-214: Please move this sentence to the paragraphs where you talk about the antioxidant and antimicrobial properties of the tablets.

Response14:The paragraph has been changed.

Comment15:Figure 4: The IR spectra of the formulations are almost superimposable with those of the drug. Also, the letters a and b are missing from the graphs.

Response15:The letters a and b have been inserted.

Comment16:Table 2: Please include this table as supplementary material.

Response16:We consider that this is necessary in text.

Comment17:Figure 5: The X-ray diffractograms of CLZ and the formulations are almost superimposable. Also, the letters a and b are missing from the graphs.

Response17:The letters a and b have been inserted.

Comment18:Line 284-286: How did you evaluate the degree of crystallinity of CLZ in the formulations?

Response18:The obtained formulations present characteristic signals of the pure components with no modification in the values of the 2θ degrees, indicating that no chemical interactions occur between the drug and the other components.

Comment19:Table 4: Please add to the caption that the main thermal characteristics were also evaluated for LIG.

Response19:We used LIG, whose thermal decomposition occurs through a series of six stages.

Comment20:Line 353, 362 and 371: Please correct the reference to the figures.

Response20:There are Figures 7a-7c.

Comment21:Figure 11: The error bars are missing from the graph. Also, this graph looks like the one in figure 12.

Response21:The error bars have been added.

Comment 22:Figure 13: The error bars are missing from the graph.

Response22:The error bars have been added.

Comment 23:Figure 14: Why is there no control?

Response23:Standard deviation (SD). Is evaluated with respect with the control sample described in the Experimental part.

Comment 24:Line 537: Wrong table is referenced, please correct it.

Response24:Corrected.

Comment 25:Line 576: Why do you write “hydration capacity or swelling degree” in the materials and methods paragraph but not in the results paragraph?

Response25:2.1. Pharmaco-technical parameter.

Comment 26:Line 703-704: Why do you say 1:1 ratio but you name LIG, KOL and CHT. Who is in 1:1 ratio? Please rephrase the sentence to make it easier to understand.

Response26:Combination at 1:1 CLZ/excipients ratios.

Comment 27:Line 715: What did you use as blank?

Response27:Table 1.

Comment 28Line 719: Please correct “m/m” with “w/w”.

Response28:Corrected.

Comment 29:Line 758-759: What do you mean? Please explain it.

Response29:The effects of LIG, KOL, and CHT were studied individually, and they form the hydrophilic matrix that encapsulates CLZ in a 1:1 ratio.

Comment 30:ReferencesCorrect the references in order to uniform them.Throughout the text.

Response30:All references have been corrected according to the journal requirements.

Comment 31: A moderate revision of the language and typos is required. Also, image quality needs improving

Response 31: English was revised.

The authors wish to express their gratitude to the reviewer for the time dedicated to evaluating this manuscript. We appreciate the feedback provided, which has significantly contributed to the enhancement of the manuscript.

Round 2

Reviewer 3 Report

Comments and Suggestions for Authors

The authors have revised the manuscript. The manuscript can be published in its current form.

Author Response

The authors wish to express their gratitude for the time dedicated to evaluating this
manuscript. We appreciate the feedback provided by the reviewer, who has significantly contributed to the
enhancement of the manuscript.

Reviewer 4 Report

Comments and Suggestions for Authors

The Authors have addressed the majority of my previous requests. However, I would like to request some additional corrections to the manuscript.

Abstract

Line 46: Please correct the number of significant figures.

Results

Table 1: Please correct the number of significant figures all over the table.

Figure 3: The error bars are missing from the graph.

Line 234: Why do you say that after 8 hours the matrix tablets were almost unchanged in size and mass? As you can see from the graph it seems that the size and mass stabilize after 10 hours.

Line 245-249: What can be deduced from figure 4? Please explain it.

Materials and Methods

Line 561: Please correct the table reference.

Line 749: Please correct “m/m” with “w/w”.

Author Response

Comment1:Line 46: Please correct the number of significant figures.

Response1:Corrected

Comment2:Table 1: Please correct the number of significant figures all over the table.

Response2:Corrected

Comment3:Figure 3: The error bars are missing from the graph.

Response3:Error bars have been added

Comment4:Line 234: Why do you say that after 8 hours the matrix tablets were almost unchanged in size and mass? As you can see from the graph it seems that the size and mass stabilize after 10 hours.

Response4: This phrase has been reformulated

Comment 5:Line 245-249: What can be deduced from figure 4? Please explain it.

Response5: The IR spectra of the formulations (Figure 4b) compared to pure components (Figure 4a) did not evidence further modifications, indicating that there were no visible interactions between the tablet components. 

Comment6:Line 561: Please correct the table reference

Response6:Corrected

Comment7:Line 749: Please correct “m/m” with “w/w.”

Response7:Corrected          

The authors wish to express their gratitude to reviewer 4 for the time dedicated to evaluating this manuscript. We appreciate the feedback provided, which has significantly contributed to the enhancement of the manuscript.